# Sequence-context-aware decoding enables robust reconstruction of protein dynamics from crystallographic B-factors

**Yiquan Wang**[1,2,*], **Minnuo Cai**[1,2], **Yahui Ma**[1], **& Kai Wei**[1,*]
[1]Xinjiang University    [2]Shenzhen X-Institute
[*]Corresponding authors
ethan@stu.xju.edu.cn   kaiwei@xju.edu.cn

## Abstract

While X-ray crystallography remains the primary source of protein structures, the B-factors associated with these coordinates are frequently obscured by crystal packing and refinement artifacts that limit their utility for quantifying solution-state dynamics. We address the disparity between static structural abundance and dynamic information scarcity by introducing the B-Factor Corrector. This fine-tuned protein language model redefines B-factor analysis as a sequence-to-dynamics translation task rather than a purely physical calculation. By leveraging deep contextual embeddings to decouple intrinsic flexibility from lattice constraints, our model recovers ground-truth conformational fluctuations derived from structural ensembles and achieves a Pearson correlation of 0.80. We further demonstrate the utility of these sequence-inferred dynamics in structural biology applications where the model identifies binding-competent conformations for flexible molecular docking, unmasks cryptic mechanical hinges essential for chaperone function, and guides the energetic refinement of antibody interfaces to resolve steric conflicts. These results suggest that static crystallographic data can be effectively repurposed to decode accurate dynamic patterns latent within the Protein Data Bank.

## Introduction

Protein function is fundamentally dynamic, relying on conformational ensembles that extend beyond static coordinates (Frauenfelder et al., 1991; Csermely et al., 2010; Nussinov et al., 2023; Fenwick et al., 2014). While X-ray crystallography has contributed over 190,000 structures to the Protein Data Bank (PDB) (Berman et al., 2000), these models are traditionally viewed as static snapshots (DePristo et al., 2004). The atomic displacement parameter, or B-factor, provides the primary experimental metric for motion; however, its interpretation remains challenging. Influenced by non-biological factors such as crystal packing contacts, solvent conditions, and refinement heterogeneity, B-factors are frequently regarded as qualitative indicators of local disorder rather than quantitative measures of solution-state dynamics (Li & Brüschweiler, 2009; Carugo, 2022; Klyshko et al., 2024; Eyal et al., 2005).

To bridge the gap between static structures and dynamic function, researchers typically rely on Molecular Dynamics (MD) simulations (Karplus & McCammon, 2002) or NMR spectroscopy (Lasorsa & van Der Wel, 2025). While accurate, these methods are computationally expensive or experimentally demanding, leaving the vast majority of PDB entries without quantitative dynamic profiling. Analytical approaches like Gaussian Network Models (GNM) (Bahar et al., 1997) and Anisotropic Network Models (ANM) (Atilgan et al., 2001) offer rapid approximations but, as they rely on coarse-grained contact topology while treating amino acids as uniform mechanical nodes, often overlook the subtle chemical specificity governed by the local sequence environment.

We postulate that the discrepancy between crystallographic B-factors and solution-state dynamics represents a systematic artifact encoded in the local physicochemical context rather than random noise. This hypothesis is supported by preliminary validations showing strong correlations between

B-factors and MD-derived RMSF on limited datasets (Wang et al., 2025). Protein sequences likely store an evolutionary memory of structural plasticity that persists even when the physical coordinate is constrained by the crystal lattice. We explicitly test this hypothesis by developing the B-Factor Corrector (BFC), a framework that leverages the deep contextual embeddings of the ESM-2 protein language model (Lin et al., 2023). Unlike traditional physical models, BFC treats the recovery of dynamics as a sequence-context-aware translation task that filters out crystal-induced rigidity to reveal the underlying solution-state fluctuations. Our results indicate that static crystallographic data, when interpreted through the lens of a language model, contains latent signals governing protein motion. Beyond reconstructing flexibility profiles, we further show that BFC serves as an effective physical prior for computational structural biology. We demonstrate that integrating these sequence-predicted dynamics into docking protocols and molecular dynamics simulations significantly improves the modeling of ligand-induced conformational changes, the identification of cryptic functional motions, and the energetic refinement of designed protein interfaces.

## RESULTS

### SEQUENCE CONTEXT ENABLES ROBUST RECONSTRUCTION OF RMSF PROFILES

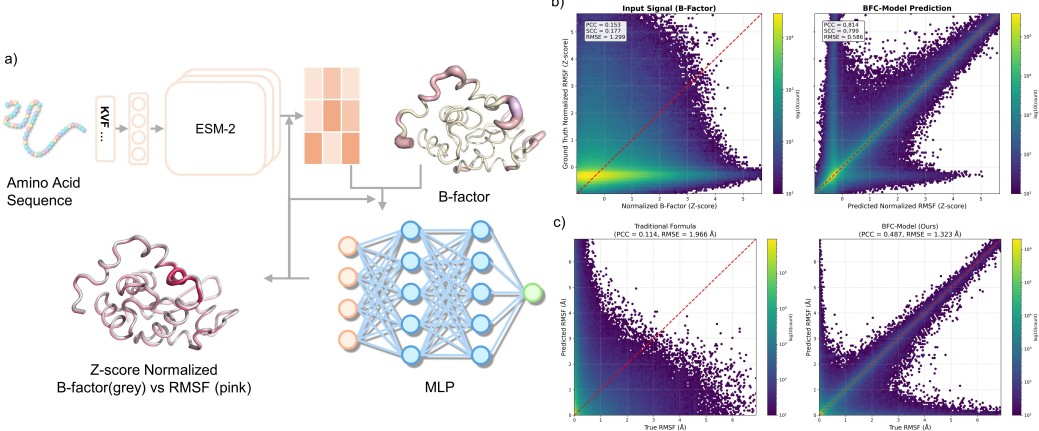

Figure 1: **Sequence-context-aware decoding enables quantitative recovery of RMSF from crystallographic B-factors. a,** Schematic of the B-Factor Corrector (BFC) workflow. The model extracts deep sequence embeddings via a fine-tuned ESM-2 and combines them with experimental B-factors to predict intrinsic dynamics. The structural visualizations illustrate the data transformation: the top structure displays the raw input B-factor distribution, while the bottom superposition reveals the discrepancy between the normalized input B-factors (grey tube) and the ground-truth ensemble RMSF (pink gradient). **b, c,** Global performance evaluation on the test set. Density scatter plots compare the correlation of predictions against ground-truth ensemble RMSF values. **b,** In normalized space (Z-score), BFC (right) effectively denoises the input B-factor signal (left), recovering a high correlation (PCC=0.80). **c,** Evaluation of profile reconstruction in physical units (Å). To isolate profile fidelity from experimental amplitude variance, both the Traditional Formula (left) and BFC (right) were rescaled to match the ground-truth moments (Oracle Rescaling). Even with correct global scaling, the traditional formula fails to capture the relative fluctuation ranking (SCC=0.25), whereas BFC accurately reconstructs the physical landscape (SCC=0.80), establishing a linear correlation with solution-state reality absent in raw data.

We first investigated whether a language model could learn the mapping between noisy crystallographic data and solution-state behavior. We benchmarked BFC on a comprehensive dataset of proteins containing both crystal structures and corresponding structural ensemble profiles derived from PDBFlex (Hrabe et al., 2016). Comparison with these ensemble-based ground truths highlighted discrepancies in the raw crystallographic data. Our analysis showed that normalized experimental B-factors exhibited a low correlation with the ensemble RMSF (Pearson Correlation Coefficient, PCC = 0.15) in the test set (fig. 1b). This suggests that raw B-factors, often influenced by lattice contacts, may diverge significantly from intrinsic flexibility.

In contrast, BFC recovered a signal consistent with the consensus flexibility of the structural ensembles, yielding a PCC of 0.80 on the test set (fig. 1b). We employed an oracle rescaling strategy to isolate profile fidelity from experimental amplitude variance and rigorously assess the recovery of physical magnitudes. This protocol standardizes both baseline and model predictions to match the first and second moments of the ground-truth ensemble and thereby focuses the evaluation on the shape of the fluctuation landscape. Even with this global scaling correction, the traditional physical formula failed to reconstruct the relative distribution (PCC = 0.11, SCC = 0.25) and indicated that the raw B-factor pattern remains fundamentally distorted. Conversely, BFC achieved a robust reconstruction of the residue-wise flexibility profile (PCC = 0.49, SCC = 0.80) and a significantly lower Mean Absolute Error (0.12 Å vs. 0.56 Å, fig. 2b). These results suggest that BFC effectively decodes the intrinsic dynamic sequence pattern latent within the static crystal structure, correcting local distortions that analytical formulas cannot address.

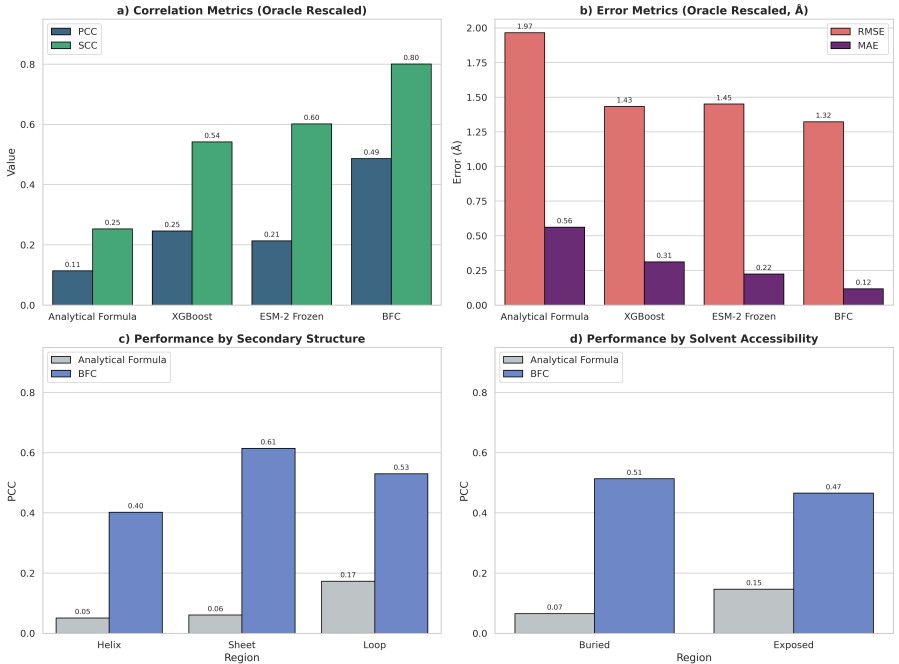

Figure 2: **BFC outperforms baselines and demonstrates robust generalization. a, b,** Systematic benchmarking of dynamic prediction accuracy. BFC (fine-tuned) significantly outperforms the analytical Gaussian Network Model (GNM), XGBoost, and pre-trained ESM-2 (frozen) in both correlation metrics (PCC/SCC) and error metrics (RMSE/MAE). The performance gap highlights the necessity of end-to-end task adaptation. **c, d,** Performance stratified by structural context. BFC maintains robust performance across secondary structures (c) and solvent accessibility levels (d), achieving high accuracy even in flexible loops and solvent-exposed regions where traditional physical formulas typically fail.

## END-TO-END FINE-TUNING CAPTURES BIOPHYSICAL NUANCES

To understand the biophysical basis of this performance, we isolated the contribution of sequence context by comparing BFC against baseline models (fig. 2a,b). A gradient-boosted decision tree (XGBoost (Chen & Guestrin, 2016)) trained solely on B-factor values did not improve correlation significantly. This observation implies that the relationship between B-factors and RMSF is likely not a simple non-linear mapping and suggests that experimental data alone, without sequence context, may be insufficient for denoising. Furthermore, we compared BFC (fully fine-tuned) with a frozen ESM-2 model, where only the output head was trained. The frozen model yielded limited improvements, suggesting that general-purpose evolutionary representations may not be inherently aligned with specific physical dynamics.

The significant performance gap achieved by end-to-end fine-tuning suggests that the model adapts its attention mechanisms to encode sequence-dependent physical properties. To confirm that this performance stems from genuine residue-wise precision rather than global scaling artifacts, we stratified the evaluation by local structural context. Detailed stratification reveals that BFC effectively learns distinct dynamic signatures at the residue level. Although buried residues exhibit slightly lower baseline fluctuations (mean Z-score: -0.05) compared to solvent-exposed regions (mean Z-score: 0.06), the model maintains high predictive accuracy within both regimes. It maintains high accuracy in buried hydrophobic cores (fig. 2d), likely by recognizing the stability conferred by high packing density. Conversely, in solvent-exposed loops, typically enriched with flexible residues like Glycine and Proline, the model achieves its most dramatic improvement over analytical baselines (fig. 2c).

Residue-level diagnostic analysis elucidates the physical logic governing the model. Isolating residues where BFC predicts high flexibility contrary to rigid experimental signals reveals that these sites comprise helices and sheets rather than disordered loops (fig. S5). This observation implies that BFC identifies metastable secondary structures that adopt a folded conformation within the static crystal lattice while possessing a sequence context prone to local unfolding. The model therefore effectively recovers latent solution-state dynamics masked by crystallization by capturing sequence-encoded intrinsic propensities beyond lattice constraints.

DENOISING CRYSTAL PACKING ARTIFACTS IN ANTIBODY CDR LOOPS

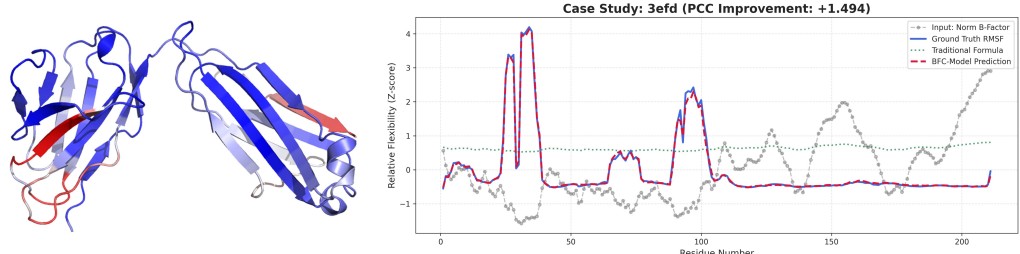

Figure 3: **Correction of crystal packing artifacts in antibody CDR loops (PDB: 3efd).** Comparison of normalized dynamic profiles reveals a major discrepancy in the CDR loop regions. Experimental B-factors (gray dotted line) falsely suggest rigidity due to crystal contacts. BFC (red dashed line) utilizes sequence context to identify these artifacts, recovering the high flexibility profile that matches the ground-truth ensemble RMSF (blue solid line). The structural map (left) visualizes the predicted flexible regions (red) versus rigid scaffolds (blue).

The biological imperative for this decoding approach is best illustrated by its ability to identify and correct misleading structural signals within our test set. In the crystal structure of an antibody fragment (PDB: 3efd (Uysal et al., 2009)), crystal packing interactions artificially stabilize the Complementarity-Determining Region (CDR) loops, leading to low B-factors that falsely suggest rigidity (Spoendlin et al., 2025; Fernández-Quintero et al., 2021). A naive interpretation of the B-factors would fail to capture the functional plasticity required for antigen binding (Blackler et al., 2022; Fernández-Quintero et al., 2019; Liu et al., 2024).

Guided by the sequence context of the CDR loops, BFC identified the experimental rigidity as a potential artifact. The model predicted a high-flexibility profile that aligned with the ground-truth ensemble variability, effectively correcting the discrepancy in the input signal (fig. 3). The increase in correlation for this specific case (+1.494 PCC) suggests that BFC does not merely smooth input data but captures sequence-dependent physical properties. The model appears to weigh sequence-encoded intrinsic propensities against experimentally observed lattice constraints, helping to recover solution-state dynamics masked by the crystallization process.

DECIPHERING DISEASE-CRITICAL DYNAMICS LATENT IN THE PDB

To validate the utility of our model for biological discovery, we applied BFC to the ATLAS database (Vander Meersche et al., 2024), an external repository of high-quality molecular dynamics simulations. Crucially, this dataset is distinct from our ensemble-based training distribution, allowing us

to test whether the model generalizes from static structural clusters to time-resolved dynamic trajectories. We analyzed the SH2 domain of SLAM-associated protein (PDB: 1D4T (Poy et al., 1999)), a critical signaling module. Consistent with the functional nature of this region, the experimental B-factors for the loop spanning residues 68–73 exhibited a distinct peak of flexibility (fig. 4c).

BFC accurately reproduced this localized peak of flexibility. Structural analysis indicates that this loop acts as a gatekeeper for the binding of the SLAMF1 ligand (fig. 4e). Notably, the Arg68 residue, located at the apex of this predicted flexible region, is the site of a missense mutation associated with X-linked lymphoproliferative syndrome type 1 (XLP1) (Booth et al., 2011). Statistical analysis confirmed that the BFC-predicted flexibility for this loop was significantly higher than the scaffold background ($p < 0.001$), identifying a functional hotspot and validating the model's capacity to capture disease-relevant dynamics from sequence context alone (fig. 4b). This capacity to pinpoint function-critical dynamics suggests that BFC can serve as a valuable tool for re-analyzing existing PDB entries to generate hypotheses regarding protein function and disease mechanisms.

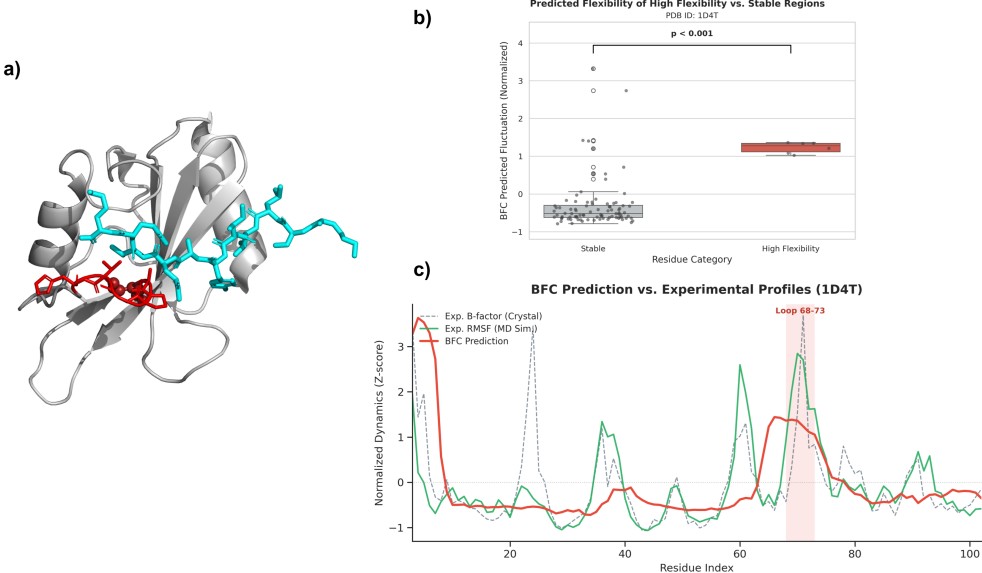

Figure 4: **Identification of disease-associated flexible motifs (PDB: 1D4T). a,** Structural visualization of the SH2 domain where BFC identifies a highly flexible loop (residues 68-73, red sticks) accommodating the SLAMF1 ligand (cyan). The Arg68 residue (red spheres) associated with XLP1 disease resides within this flexible region. **b,** Statistical validation indicating that BFC-predicted flexibility for the binding loop is significantly elevated compared to the stable regions ($p < 0.001$). **c,** Comparative dynamic profiling against external Molecular Dynamics simulations. The BFC prediction (red solid line) is overlaid with experimental crystallographic B-factors (grey dashed line) and ground-truth RMSF derived from the ATLAS MD database (green solid line). BFC accurately recovers the solution-state flexibility peak at the binding loop (residues 68-73) consistent with the MD trajectory, confirming the capacity of the model to generalize from structural ensembles to time-resolved simulations and pinpoint pathogenicity-relevant dynamics.

UNMASKING CRYPTIC MECHANICAL HINGES LATENT IN STATIC STRUCTURES

While the previous case demonstrated consistency between BFC and molecular dynamics, a more profound application lies in identifying functional motions that evade detection by crystallography and standard equilibrium simulations. We analyzed the Rubisco accumulation factor 1 (RAF1, PDB: 4WT3) as a critical chaperone for plant photosynthesis (Hauser et al., 2015). Standard structural metrics present a limited view in which both crystallographic B-factors and ATLAS MD simulations depict the $\alpha$-domain as a rigid scaffold (Fig. 5a). BFC conversely predicts a solitary and high-intensity flexibility peak centered on Helix $\alpha3$ (residues 107–124).

We resolved this discrepancy by performing a sequence alignment between the crystallized *A. thaliana* RAF1 and its functional homolog from *S. elongatus* (Syn7942). This analysis reveals that the flexible region predicted by BFC corresponds to Helix $\alpha3$ and harbors a highly conserved Glycine residue (G112). This residue acts as a helix destabilizer to impart an intrinsic potential for local unfolding that the crystal lattice suppresses (Fig. 5b,c).

Evolutionary mapping of cross-linking mass spectrometry (CXMS) data (Hauser et al., 2015) resolves the mechanistic logic of this prediction (Fig. 5d,e). Experimental evidence establishes that RAF1 anchors to its massive Rubisco substrate via two flanking contact epitopes comprising Lys29 in the upstream Helix $\alpha2$ and Lys78 in the downstream Helix $\alpha4$. This topology creates a structural paradox in the rigid crystal model because the simultaneous grasping of the curved Rubisco surface by these distinct upstream and downstream extensions requires the intervening Helix $\alpha3$ to function as a conformational hinge. BFC successfully decodes the sequence-encoded plasticity required for this clamping mechanism and identifies a functional dynamic feature that permits induced-fit recognition despite the frozen state of the apo crystal.

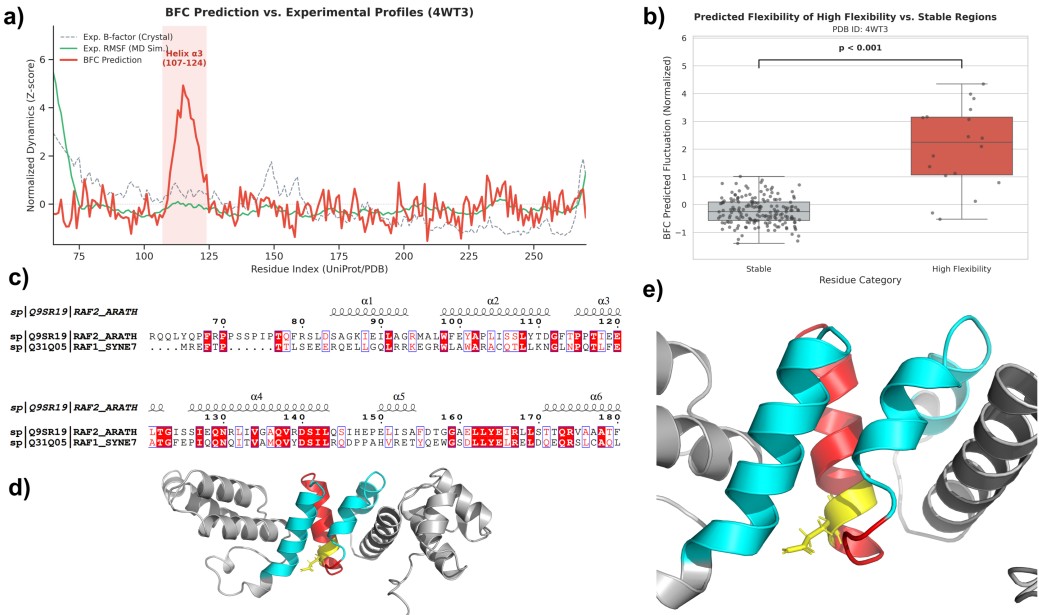

Figure 5: **BFC unmasks a cryptic mechanical hinge in the Rubisco chaperone RAF1 (PDB: 4WT3). a,** Comparison of structural metrics where crystallographic B-factors (grey dashed) and standard MD simulations (ATLAS database, green) depict the $\alpha$-domain as uniformly rigid. Conversely, BFC (red) predicts a distinct high-flexibility peak corresponding to Helix $\alpha3$ (residues 107–124). **b,** Statistical analysis confirming that the predicted fluctuation of this region is significantly higher than the stable background of the protein ($p < 0.001$). **c,** Sequence alignment with the homolog *S. elongatus* Raf1 reveals the conserved nature of the predicted region where a conserved Glycine residue (red box) acts as a structural pivot within the helix. **d, e,** Structural mapping of the hinge mechanism. **d,** Overview of the RAF1 monomer. **e,** Zoomed view showing the mechanical architecture where the BFC-predicted flexible region corresponds to the central Helix $\alpha3$ (red) bridging two rigid anchor helices $\alpha2$ and $\alpha4$ (cyan). The conserved Glycine residue (G112, yellow sticks) provides the pivotal freedom required for the red helix to bend. This arrangement enables the rigid cyan anchors to reorient for substrate clamping which is a dynamic feature essential for chaperone function but masked in the static crystal structure.

APPLICATION TO MOLECULAR DOCKING AND INTERFACE REFINEMENT

To demonstrate that BFC-predicted flexibility constitutes a functional asset rather than merely a statistical correlate, we integrated the model into two critical computational structural biology tasks: flexible molecular docking and antibody interface refinement.

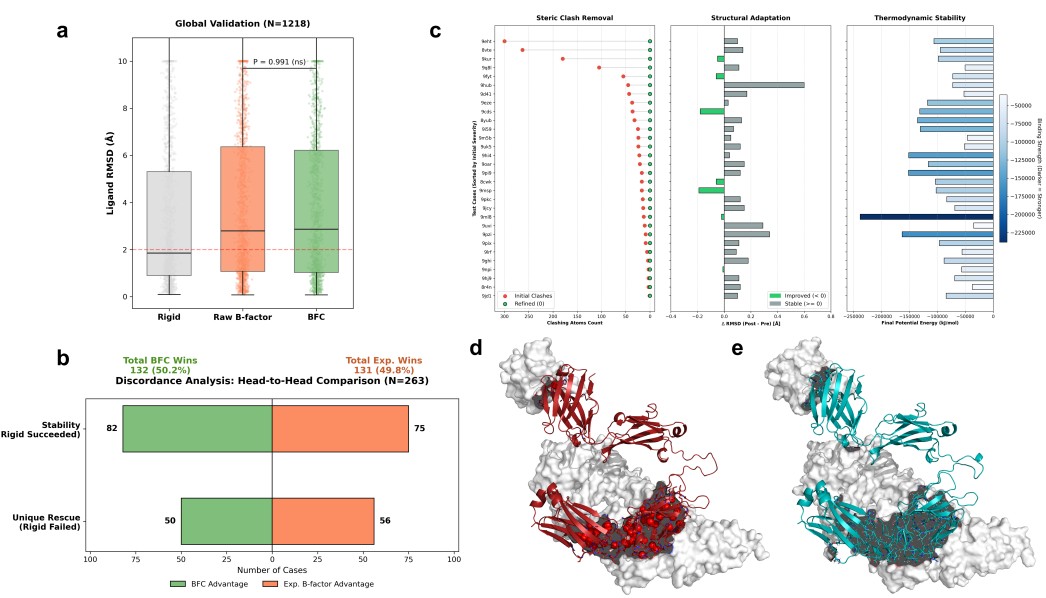

Figure 6: **Utility of sequence-derived flexibility for molecular docking and antibody interface refinement. a,** Performance parity in molecular docking simulations (N=1,218). Boxplots comparing ligand RMSD distributions on the PDBbind refined set show that docking guided by BFC-predicted flexibility (Green) yields results statistically indistinguishable from those guided by experimental crystallographic B-factors (Orange, $P = 0.991$, Wilcoxon signed-rank test) while both methods significantly expand the sampling space compared to the rigid baseline (Grey). **b,** Resolution of crystallographic artifacts (N=263). Discordance analysis of conflicting outcomes reveals that BFC successfully rescued 50 targets where both rigid and experimental B-factor protocols failed (Green region), suggesting that the model effectively filters non-biological artifacts to recover functional flexibility. **c,** Energetic optimization via molecular dynamics (N=30). Performance metrics for 30 antibody-antigen complexes refined using BFC-guided dynamics demonstrate that the protocol resolves severe steric clashes (Red to Green transition) and drives the system from high-energy conflicts to stable potential energies ($\Delta E$) while maintaining structural integrity ($\Delta$RMSD). **d-e,** Structural mechanism of flexibility-guided induced fit for PDB 9EHT. **d,** The initial state derived from rigid alignment of the ESMFold prediction (Dark Red) to the antigen (White) exhibits severe steric clashes marked by red spheres that correspond to a physically invalid high-energy state. **e,** The final refined structure (Teal) obtained after applying BFC-guided restraints shows conformational adaptation of the antibody CDR loops to the antigen surface that fully resolves steric clashes and achieves high shape complementarity.

We addressed the limitations of rigid-receptor docking through a comprehensive benchmark on the PDBbind Refined Set comprising 1,218 complexes (Wang et al., 2005; Liu et al., 2017). Traditional protocols often fail to accommodate ligand-induced conformational changes and the use of raw crystallographic B-factors to identify flexible residues risks introducing lattice artifacts. We compared docking outcomes guided by BFC predictions against those guided by experimental B-factors and observed global physical equivalence. The distribution of successful ligand RMSD values derived from BFC guidance proved statistically indistinguishable from that derived from experimental data ($P = 0.991$, Wilcoxon signed-rank test; Fig. 6a). A discordance analysis of cases where the two methods diverged provided further functional validation of residue-wise accuracy. BFC successfully rescued 50 targets where both rigid docking and experimental B-factor protocols failed (RMSD > 2.0 Å; Fig. 6b). Because the docking protocol strictly limits flexibility to the top three residues, this success confirms that the model correctly pinpoints the specific subset of binding pocket side-chains that require conformational adaptation. The language model effectively filters out non-biological crystal packing constraints and identifies flexible residues that appeared artificially rigid in the crystal lattice to enable the sampling of correct induced-fit conformations.

We extended the application of BFC to refine antibody-antigen interfaces derived from static sequence-based predictions. While state-of-the-art models such as ESMFold (Lin et al., 2023) generate high-quality backbones, they typically produce static and unliganded conformations. When these structures are rigidly aligned to antigens, the absence of induced fit frequently results in severe steric overlaps and physically implausible interface geometries characterized by extremely high potential energies. To address this limitation, we developed a Molecular Dynamics (MD) refinement pipeline in which BFC scores modulate differential restraint strengths by imposing strict constraints on the rigid framework while enabling the relaxation of flexible CDR loops identified by BFC. Across 30 test complexes, these flexibility-guided simulations successfully transitioned the systems from high-energy states to thermodynamically stable minima. This protocol effectively eliminated severe steric clashes and significantly reduced system potential energy (Fig. 6c). As demonstrated by the PDB 9EHT complex (Fig. 6d,e), our method converts initial states exhibiting atomic collisions into shape-complementary interfaces. By incorporating the distinct thermodynamic landscape of the antibody variable region into the simulation, BFC bridges the gap between static sequence predictions and biologically valid structural models.

## DISCUSSION

Our study challenges the prevailing view that crystallographic B-factors contain excessive noise for quantitative dynamic analysis. We demonstrate that treating B-factors as signals modulated by environmental constraints allows deep learning models to infer solution-state dynamics directly from sequence context. This approach mirrors recent breakthroughs in protein design (Dauparas et al., 2022; Watson et al., 2023; Dauparas et al., 2025; Hayes et al., 2025) where deep learning bridges the gap between sequence and physical realizability. Our results indicate that B-factor variability represents a convolution of intrinsic motion and environmental factors rather than random noise. Although BFC operates algorithmically as a data-driven statistical mapping rather than explicitly computing thermodynamic forces to decouple lattice contacts from first principles, its biological fidelity transcends naive global smoothing. The model's ability to assign function-critical flexibility on a strictly per-residue basis, even when correcting misleading experimental signals (e.g., pinpointing the cryptic G112 hinge in RAF1), demonstrates profound biophysical awareness. By treating the ESM-2 representations as an evolutionary prior, the network learns a highly localized mapping that effectively achieves a physical disentanglement of intrinsic propensity from environmental artifacts. Crucially, whereas topological models demand complete complex coordinates to define mechanical constraints, BFC operates on isolated chains by capturing environmental signatures including interface rigidification that are implicitly imprinted on the input B-factors. Although the prediction of absolute fluctuation amplitudes faces constraints from unknown global scaling factors (PCC=0.49), BFC reconstructs the relative dynamic distribution with high precision (PCC=0.80) and significantly outperforms traditional physical models.

The ability to decouple intrinsic plasticity from lattice artifacts offers immediate advantages for structural modeling. By resolving structural paradoxes where rigid crystal models contradict obligate functional motions, BFC uncovers essential mechanical features such as chaperone hinges that remain obscured even in standard equilibrium simulations. In flexible molecular docking benchmarks, the model successfully identified binding site residues that appeared rigid in the crystal lattice but required conformational adaptation for ligand accommodation. This allowed the rescue of targets that failed under experimental guidance and suggests that BFC effectively filters out false negatives in flexibility assignment. Similarly, the targeted application of BFC-derived restraints during the refinement of antibody-antigen complexes bridged the gap between static prediction and physical realizability. By permitting relaxation specifically within sequence-identified flexible loops while maintaining scaffold integrity, the protocol resolved high-energy steric conflicts that are typically associated with rigid-body alignment. These applications confirm that sequence-derived flexibility constitutes a functional asset that complements static coordinate data.

This finding opens the door to dynamic mining of the PDB, allowing researchers to revisit over 190,000 static structures and extract quantitative dynamic information without the prohibitive computational cost of molecular dynamics simulations (Jung et al., 2019; Sahil et al., 2023; Prašnikar et al., 2024; Lappala, 2024; Cui et al., 2025). Furthermore, by validating that a compact language model suffices for this translation task, we ensure that BFC remains computationally efficient and deployable for high-throughput screening on standard workstations. While BFC is not a replace-

ment for MD in studying complex transitions, it provides a computationally efficient approximation of intrinsic conformational flexibility, serving as a bridge between static structural biology and dynamic biochemistry. We anticipate that BFC will facilitate the large-scale retrospective analysis of PDB entries to investigate mechanisms of protein function and disease.

## MEANINGFULNESS STATEMENT

A meaningful representation of life must capture the inherent dynamism of biological molecules rather than limiting them to static coordinates. We contend that the evolutionary history encoded in protein sequences contains a latent blueprint of physical flexibility. Our work advances this perspective by using language models to extract solution state dynamics from crystallized structures. By decoupling intrinsic motion from experimental artifacts, we transform frozen snapshots into functional ensembles. This methodology demonstrates that understanding life requires decoding the fluctuating energy landscapes that govern molecular interaction, ultimately bridging the gap between static structural biology and dynamic physiology.

## ACKNOWLEDGMENTS

We are deeply grateful to Prof. Tieliu Shi for his invaluable guidance and contributions to this work. This work was supported in part by the Computing and Data Center of Xinjiang University. We acknowledge the computing resources and technical support provided by the Computing and Data Center of Xinjiang University.

## FUNDING

This work was supported by the Key Research and Development Project of Xinjiang Autonomous Region, China (Grant No. 2025B02008-1), the National Natural Science Foundation of China (Grant No. 32500528), the Natural Science Foundation of Xinjiang Uygur Autonomous Region (Grant No. 2024D01C216), and the "Tianchi Talents" introduction plan.

## DATA AND CODE AVAILABILITY

Source code for BFC and the pre-trained model weights are freely available at `https://github.com/wyqmath/BFactorCorrector`. The raw data used in this study were obtained from the publicly available RCSB Protein Data Bank (PDB) and PDBFlex database. External validation was performed using data from the ATLAS database. Pre-processed training datasets and other supporting data are available from Yiquan Wang (ethan@stu.xju.edu.cn) upon reasonable request.

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

## A  APPENDIX

### METHODS

#### DATASET CONSTRUCTION AND PREPROCESSING

To rigorously test the hypothesis that solution-state dynamics can be quantitatively recovered from crystallographic data, we constructed a large-scale, high-fidelity benchmark connecting static PDB structures with dynamic MD ensembles.

**Data Acquisition and Filtering.** We retrieved high-resolution X-ray structures from the RCSB Protein Data Bank (PDB) and matched them with ground-truth Root Mean Square Fluctuation (RMSF) profiles derived from structural ensembles in the PDBFlex database (Hrabe et al., 2016). Specifically, ground-truth RMSF values were calculated based on the coordinate variance of $C\alpha$ atoms across the aligned ensemble members to capture backbone flexibility. To ensure that the model learns from reliable physical signals rather than noise, we applied a strict quality control pipeline: (1) *Experimental Constraints*: Beyond relying on database metadata, we explicitly parsed PDB headers (specifically EXPDTA records) to confirm X-ray diffraction as the method. Only structures with a resolution $\leq 3.0$ Å were retained. (2) *Sequence Processing and Alignment*: A strict residue-level mapping was enforced. Non-standard Selenomethionine (MSE) residues were mapped to Methionine (MET) to handle common crystallographic phasing derivatives, while other non-standard residues were treated as gaps. Chains with significant missing electron density (gaps $> 10\%$) or mismatches between $C\alpha$ coordinates and the sequence registry were discarded. (3) *Statistical Quality Control*: To filter out corrupted data entries or flat-lined profiles, we calculated the standard deviation of input B-factors and ground-truth RMSF for each chain. Entries exhibiting effectively zero variance ($\sigma < 10^{-6}$) were considered artifactual and removed. (4) *Context Sufficiency*: Peptide chains shorter than 50 residues were excluded to ensure sufficient sequence context for the language model. The final curated dataset comprises 280,487 protein chains derived from 132,386 unique PDB entries.

**Input Representation and Normalization.** Consistent with our goal of decoding intrinsic signals solely from crystallographic data, we minimized the use of hand-crafted features. The model inputs are restricted to two modalities: the raw amino acid sequence (providing physicochemical context) and the experimental B-factors (providing the latent dynamic signal). To address the variability in B-factor magnitudes caused by differing refinement protocols and crystal packing densities across the PDB, we standardized the input B-factors ($B_i$) and target RMSF values ($R_i$) using chain-specific Z-score normalization:

$$x_i' = \frac{x_i - \mu_{chain}}{\sigma_{chain}} \tag{1}$$

where $\mu_{chain}$ and $\sigma_{chain}$ denote the mean and standard deviation for the chain. This normalization compels the model to focus on the relative distribution of fluctuations along the sequence rather than absolute global magnitudes, which are often artifactual.

**Data Partitioning.** To prevent data leakage and ensure that the model generalizes to novel protein folds, the dataset was split at the PDB-entry level. The 132,386 unique entries were randomly partitioned into training (207,465 chains), validation (52,268 chains), and testing (20,754 chains) sets. This strategy ensures that homologous chains derived from the same crystal structure do not appear simultaneously in training and evaluation partitions.

FEATURE ENGINEERING AND INPUT REPRESENTATION

**Minimalist Sequence-Only Input Strategy.** To rigorously evaluate the hypothesis that the solution-state dynamics can be decoded directly from the interplay between evolutionary constraints and experimental noise, we adopted a minimalist input strategy. We deliberately excluded explicit structural descriptors derived from coordinate geometry (such as secondary structure assignments, solvent accessibility, or contact maps) and theoretical computations (such as Normal Mode Analysis). The model relies strictly on two input modalities:

- **Sequence Embeddings:** The raw amino acid sequence of each individual chain serves as the sole source of physicochemical and evolutionary context. Sequences were tokenized and processed by the pre-trained ESM-2 language model (`esm2_t6_8M_UR50D`). We specifically selected this efficient 8-million parameter architecture to ensure that the inference framework remains accessible on standard consumer hardware while leveraging the state-of-the-art evolutionary extraction capabilities of the ESM family to generate token-level representations $\mathbf{h}_i \in \mathbb{R}^{320}$.

- **Normalized B-Factors:** The crystallographic B-factor is provided as the only experimental observation. To normalize the vast differences in refinement scales and crystal qualities across the PDB, raw B-factors were standardized using chain-level Z-score normalization prior to input:

$$B'_i = \frac{B_i - \mu_{chain}}{\sigma_{chain}} \tag{2}$$

This architecture treats B-factors as noisy proxies for dynamics, leveraging sequence context to denoise and translate these signals into solution-state RMSF without relying on engineered physical features.

B-FACTOR CORRECTOR (BFC) ARCHITECTURE

BFC treats dynamics prediction as a token-level regression task. The architecture consists of the ESM-2 encoder followed by a specialized decoding head.

**Encoder.** We employed the ESM-2 transformer as a feature extractor. Unlike approaches that freeze the language model, we performed full parameter fine-tuning on the ESM-2 backbone. This allows the attention mechanisms to adapt specifically to the task of recognizing crystallographic artifacts and correlating them with sequence motifs.

**Prediction Head.** The embeddings $\mathbf{h}_i$ are concatenated with the normalized B-factor scalar $B'_i$. This composite vector is passed through a regression head consisting of a linear projection (dim=256), Layer Normalization, and a ReLU activation function. A Dropout layer ($p = 0.1$) is applied for regularization before the final linear projection to a scalar value representing the predicted RMSF.

BASELINE MODELS

We benchmarked BFC against three distinct classes of models to isolate the source of performance gains:

**Analytical Formula:** A traditional physical conversion where RMSF is derived directly from B-factors assuming an isotropic harmonic oscillator (Kuzmanic & Zagrovic, 2010):

$$\text{RMSF} = \sqrt{\frac{3B}{8\pi^2}} \tag{3}$$

**XGBoost (B-Factor Only):** To determine if the relationship between B-factors and RMSF is simply a non-linear mapping independent of sequence context, we trained a gradient-boosted decision tree regressor (XGBoost (Chen & Guestrin, 2016)) using *only* the normalized B-factors as input features. Hyperparameters (max_depth=5, n_estimators=1000) were optimized on the validation set.

**Frozen ESM-2:** To assess the necessity of fine-tuning, we trained the same prediction head described above on top of fixed embeddings from the pre-trained ESM-2 model, without updating the transformer weights.

TRAINING AND OPTIMIZATION

**Loss Function.** We minimized the Mean Absolute Error (MAE) between the predicted and ground-truth RMSF values. Crucially, to handle variable sequence lengths within batches, we implemented a masking mechanism where padding tokens (assigned a placeholder label of -1.0) are strictly excluded from the computation. The loss is calculated as:

$$\mathcal{L} = \frac{1}{N_{valid}} \sum_{i=1}^{N_{total}} m_i \cdot |y_i - \hat{y}_i| \tag{4}$$

where $m_i \in \{0, 1\}$ is the binary mask indicating valid residues, and $N_{valid}$ is the total number of non-padding tokens in the batch.

**Optimization.** Training was performed using the AdamW optimizer (Loshchilov & Hutter, 2019) with a learning rate of $5 \times 10^{-5}$ and a batch size of 64. We employed a linear learning rate scheduler with warmup, where the learning rate increases linearly for the first 10% of training steps and decays linearly thereafter.

**Implementation.** All models were implemented in PyTorch 2.6.0 (Paszke, A., Gross, S., Massa, F., Lerer, A., Bradbury, J., Chanan, G., ... & Chintala, S., 2019) and trained on a single NVIDIA GeForce RTX 5090 (CUDA 13.0). To balance computational efficiency with data integrity, input sequences were truncated to a maximum length of 1,024 residues. Statistical analysis confirms that this threshold covers 99.37% of the dataset (Supplementary Figure S5), ensuring negligible information loss. Training proceeded for a fixed duration of 100 epochs, and the checkpoint achieving the highest Pearson Correlation Coefficient (PCC) on the validation set was selected for testing.

EVALUATION METRICS

To comprehensively assess model performance across both relative fluctuation patterns and absolute physical magnitudes, we employed four metrics. All metrics were calculated individually for each protein chain $k$ with length $L_k$, and then averaged over the total number of chains in the test set ($N$). Let $\mathbf{y} = (y_1, \ldots, y_{L_k})$ denote the predicted RMSF profile and $\hat{\mathbf{y}} = (\hat{y}_1, \ldots, \hat{y}_{L_k})$ denote the ground-truth ensemble RMSF.

**Pearson Correlation Coefficient (PCC):** Measures the linear correlation between the predicted and experimental profiles, assessing the model's ability to capture the shape of the fluctuation landscape.

$$\text{PCC} = \frac{1}{N} \sum_{k=1}^{N} \frac{\sum_{i=1}^{L_k} (y_i - \bar{y})(\hat{y}_i - \bar{\hat{y}})}{\sqrt{\sum_{i=1}^{L_k} (y_i - \bar{y})^2} \sqrt{\sum_{i=1}^{L_k} (\hat{y}_i - \bar{\hat{y}})^2}} \tag{5}$$

**Spearman Correlation Coefficient (SCC):** Evaluates the monotonic rank-order correlation. This metric is robust to outliers and assesses whether the model correctly ranks residues from most rigid to most flexible.

$$\text{SCC} = \frac{1}{N} \sum_{k=1}^{N} \left( 1 - \frac{6 \sum_{i=1}^{L_k} d_i^2}{L_k(L_k^2 - 1)} \right) \tag{6}$$

where $d_i$ is the difference between the ranks of $y_i$ and $\hat{y}_i$.

**Root Mean Square Error (RMSE):** Quantifies the average magnitude of error in physical units (Å). This is the primary metric for assessing quantitative accuracy, penalizing large deviations more

heavily than small ones.

$$\text{RMSE} = \frac{1}{N} \sum_{k=1}^{N} \sqrt{\frac{1}{L_k} \sum_{i=1}^{L_k} (y_i - \hat{y}_i)^2} \tag{7}$$

**Mean Absolute Error (MAE):** Provides a linear score of error magnitude, offering a direct interpretation of the average deviation per residue.

$$\text{MAE} = \frac{1}{N} \sum_{k=1}^{N} \left( \frac{1}{L_k} \sum_{i=1}^{L_k} |y_i - \hat{y}_i| \right) \tag{8}$$

**Physical Space Reconstruction Assessment (Oracle Rescaling).** Since solution-state fluctuation amplitudes vary independently of sequence due to experimental conditions (e.g., temperature, crystal quality), direct prediction of absolute RMSF magnitudes from static coordinates is ill-posed. To decouple intrinsic profile accuracy from global amplitude scaling, we applied an oracle rescaling strategy for the physical space evaluation presented in fig. 1c. For each protein chain, both the BFC output (Z-score) and the baseline physical estimates derived from B-factors were linearly transformed to match the mean ($\mu_{GT}$) and standard deviation ($\sigma_{GT}$) of the ground-truth ensemble RMSF. This transformation ensures that the evaluation metrics (RMSE, PCC, SCC) strictly measure the fidelity of the fluctuation landscape shape (profile reconstruction) rather than systematic scaling errors.

EXTERNAL VALIDATION AND FUNCTIONAL ANNOTATION

To benchmark the biological fidelity of BFC predictions, we utilized the ATLAS database of standardized molecular dynamics simulations. Ground-truth RMSF profiles were established by computing the ensemble average of the three independent trajectory replicates provided for each entry, ensuring robustness against stochastic sampling errors.

For the functional case studies presented (e.g., PDB 1D4T and 1CUO), critical functional regions such as ligand-binding loops and active site adjacencies were identified and mapped according to the UniProt Knowledgebase (UniProt Consortium, 2018) annotations. To quantitatively verify the correspondence between predicted dynamics and biological function, we implemented a statistical scoring workflow. The BFC-predicted normalized fluctuations of these UniProt-annotated regions were compared against the remaining structural scaffold. A Welch's t-test for unequal variances was employed to assess significance, with a threshold of $p < 0.001$ confirming that the model selectively assigns significantly higher flexibility to function-critical motifs compared to the stable protein core.

MOLECULAR DOCKING BENCHMARK

**Dataset construction and preprocessing.** To establish a rigorous paired benchmark connecting sequence-derived dynamics with binding conformational changes, we curated the PDBbind Refined Set (v2020), comprising 5,316 entries. From this initial pool, we applied a strict multi-stage filtration pipeline to ensure chemical and structural consistency. Complexes exhibiting missing receptor backbones, inconsistent ligand chemistry, or failures in geometric center calculation during Meeko preparation were excluded. To maintain a strictly paired statistical design, we retained only those targets that successfully completed all three parallel docking protocols (Rigid, Experimental-guided, and BFC-guided). This rigorous intersection process reduced the initial dataset to a high-quality benchmark of 1,218 protein-ligand complexes, ensuring that performance metrics reflect genuine algorithmic differences rather than dataset heterogeneity.

**Sequence-based flexibility inference.** We utilized the fine-tuned BFC model to infer residue-level flexibility directly from the amino acid sequence. For each target chain, the raw sequence was tokenized and processed by the ESM-2 backbone to generate latent representations, which were then decoded into normalized flexibility scores (Z-scores). Crucially, this inference process relied exclusively on sequence data without accessing coordinate information or contact maps, thereby preventing the leakage of crystal packing artifacts into the flexibility prediction.

**Definition of flexible search space.** To evaluate the physical fidelity of the predicted dynamics, we implemented a controlled budget comparisons strategy. The binding pocket was defined as all

residues within a 6.0 Å radius of the crystal ligand. We compared two distinct flexibility selection protocols against a rigid baseline. In the experimental protocol, the three residues within the pocket exhibiting the highest crystallographic C$\alpha$ B-factors were designated as flexible. In the BFC protocol, the top three residues with the highest predicted flexibility scores were selected. By fixing the degree of freedom budget at $N = 3$ residues, we ensured that any divergence in docking performance was attributable solely to the quality of the flexibility identification source rather than the complexity of the conformational search space.

**Docking protocol and evaluation metrics.** Molecular docking simulations were executed using Uni-Dock (Yu et al., 2023), a GPU-accelerated implementation of the AutoDock Vina (Trott & Olson, 2010) scoring function. To ensure robust sampling of the expanded conformational landscape induced by side-chain flexibility, we employed a high exhaustiveness setting of 128 with a maximum of 20 generation steps. The search space was dynamically defined for each target, centered on the native ligand geometry with a padding of 10.0 Å in all dimensions. Simulations were executed in isolated threads to enforce deterministic reproducibility. Docking accuracy was quantified by the Root Mean Square Deviation (RMSD) of the top-ranked pose relative to the crystal reference, with a success threshold defined at $\leq 2.0$ Å. To explicitly quantify the denoising capability of the model, we performed a discordance analysis to identify rescue cases, defined as targets where both rigid and experimental B-factor guided docking failed (RMSD $> 2.0$ Å) but BFC-guided docking succeeded. This metric serves as a proxy for the ability of the model to filter out lattice-induced rigidity artifacts and recover functional plasticity.

### ANTIBODY INTERFACE REFINEMENT

**System Setup.** A diverse dataset of 30 antibody-antigen complexes was curated from SAbDab (Dunbar et al., 2014). Initial single-chain variable fragment (scFv) structures were predicted using ESMFold with a flexible linker (Lin et al., 2023). To simulate a blind docking scenario, these predictions were superposed onto the native antibody coordinates relative to the antigen using heavy chain framework C$\alpha$ atoms, intentionally preserving interface steric clashes.

**BFC-Guided Simulation.** Refinement was conducted using OpenMM (v8.4) (Eastman et al., 2023) with the Amber14 force field and GBSA-OBC2 implicit solvent (Onufriev et al., 2004). Preprocessing included pH 7.0 protonation, automatic disulfide bond reconstruction (CYS to CYX), and terminal oxygen geometry correction. We implemented a differential restraint scheme based on the BFC flexibility map (threshold $\tau = 0.5$). Strict positional restraints were applied to the antigen ($k = 500$ kJ/mol/nm$^2$) and rigid antibody frameworks ($k = 100$ kJ/mol/nm$^2$) to maintain topology, while flexible loops received minimal restraints ($k = 1.0$ kJ/mol/nm$^2$) to enable induced fit. The protocol employed an adaptive Langevin dynamics strategy: following multi-stage energy minimization, a 100 ps production run was performed at 300 K. An automated fallback mechanism activated a stable mode (reduced time step 0.5 fs, high friction $5.0$ ps$^{-1}$) upon detecting thermodynamic instability.

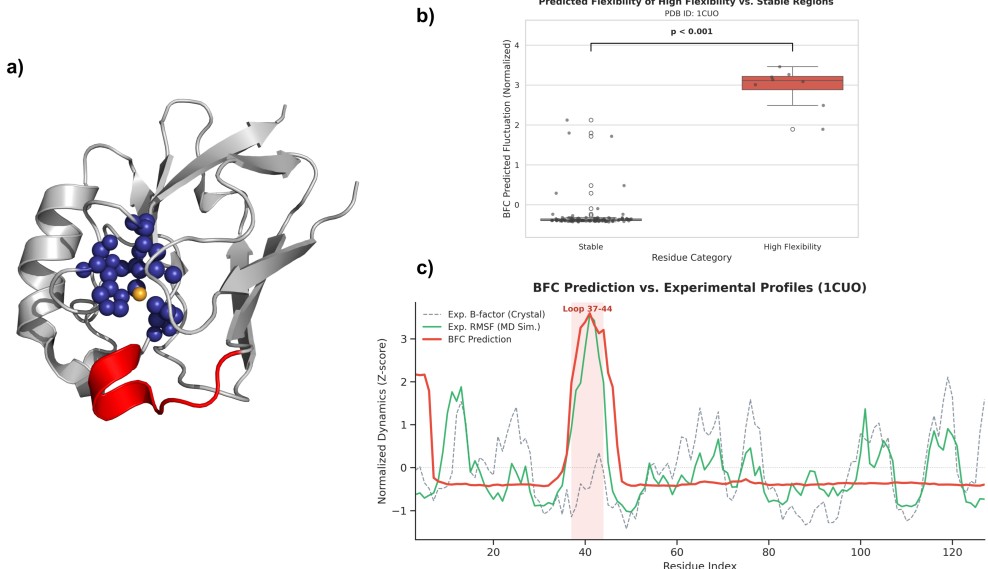

Figure S1: **BFC decodes the functional duality of rigidity and flexibility in Azurin (PDB: 1CUO). a,** Structural visualization of the Azurin active site. The landscape reveals a sophisticated dynamic duality: BFC correctly assigns structural rigidity to the core residues (blue spheres) that form a precise scaffold to coordinate the copper ion (orange). Simultaneously, the model identifies a distinct, highly flexible loop (residues 37-44, highlighted in red) adjacent to the core known to mediate electron transfer interactions. **b,** Statistical validation. A quantitative comparison shows that the BFC-predicted flexibility for the functional loop (red box) is significantly higher than that of the protein's stable regions ($p < 0.001$, Welch's t-test). **c,** Comparative dynamic profiling against external Molecular Dynamics simulations. The plot overlays the BFC prediction (red solid line) with experimental crystallographic B-factors (grey dashed line) and ground-truth RMSF derived from the ATLAS MD database (green solid line). Notably, while the crystallographic B-factors suggest a dampened motion for the functional loop (residues 37-44) likely due to lattice constraints, BFC accurately recovers the high-flexibility peak consistent with the solution-state MD trajectory. This validates the model's ability to generalize from structural ensembles to time-resolved simulations.

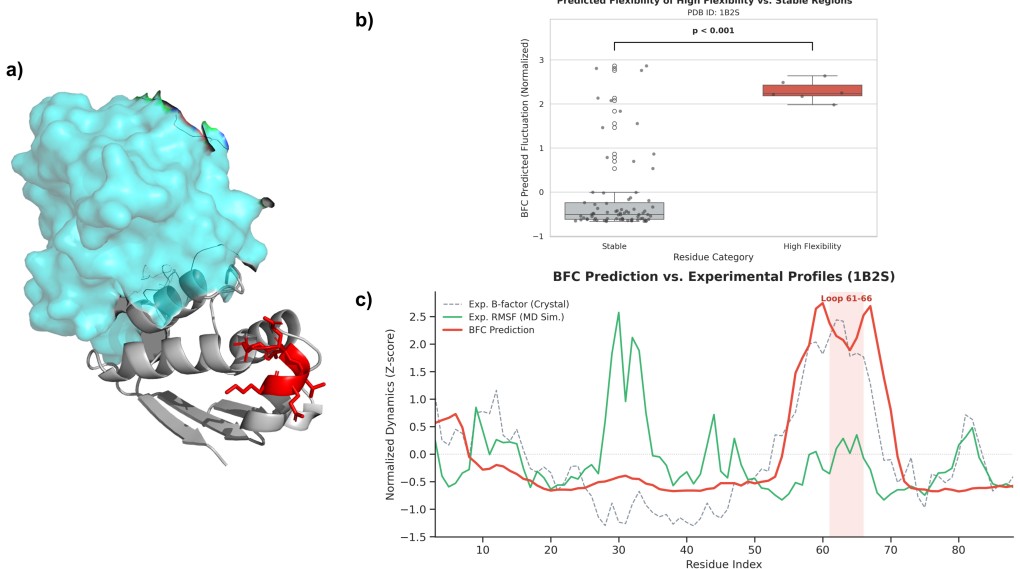

Figure S2: **BFC reveals intrinsic flexibility in Barstar (PDB: 1B2S) consistent with crystallographic evidence. a,** Structural context of the Barstar (Chain E, gray ribbon) and Barnase (cyan surface) complex. The BFC-predicted high-flexibility region (residues 61-66) is highlighted in red, identifying a solvent-exposed loop with high propensity for intrinsic motion. **b,** Statistical validation of the prediction. The BFC-predicted fluctuation scores for the identified loop (red box) are significantly higher than those of the stable scaffold ($p < 0.001$, Welch's t-test). **c,** Comparative dynamic profiling against external Molecular Dynamics simulations. The plot overlays the BFC prediction (red solid line) with experimental crystallographic B-factors (grey dashed line) and ground-truth RMSF derived from the ATLAS MD database (green solid line). In the loop region (residues 61-66, shaded red), BFC predicts a high-flexibility peak that closely aligns with the experimental B-factors, confirming the intrinsic disorder of this motif. In contrast, the ATLAS MD simulation exhibits lower fluctuations. This deviation is expected as BFC is trained on structural ensembles that capture the full conformational propensity often reflected in B-factors, whereas individual MD trajectories may sample restricted metastable states. The result validates that BFC effectively recovers sequence-encoded flexibility consistent with experimental observation.

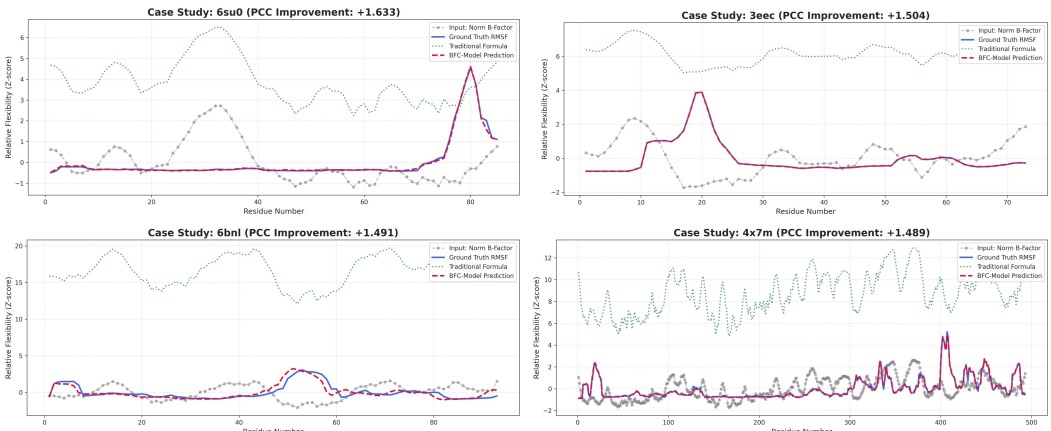

Figure S3: **Extended Case Studies of dynamic profile reconstruction.** This figure illustrates detailed flexibility predictions for four proteins with varying sequence lengths, including 6su0, 3eec, 6bnl, and 4x7m. The plots display the BFC-Model Prediction (red dashed line) in comparison to the Ground Truth RMSF (blue solid line), the Input Norm B-Factor (gray connected dots), and the Traditional Formula (green dotted line). The BFC model exhibits consistent alignment with the ground truth trajectory across all cases, effectively identifying distinct flexibility peaks in both short peptides and larger protein structures such as 4x7m. The PCC improvement scores noted in the panel headers quantify the increase in Pearson correlation coefficient achieved by the model relative to the normalized B-factor input. These metrics validate the capacity of BFC to mitigate experimental artifacts and accurately recover intrinsic biophysical dynamics.

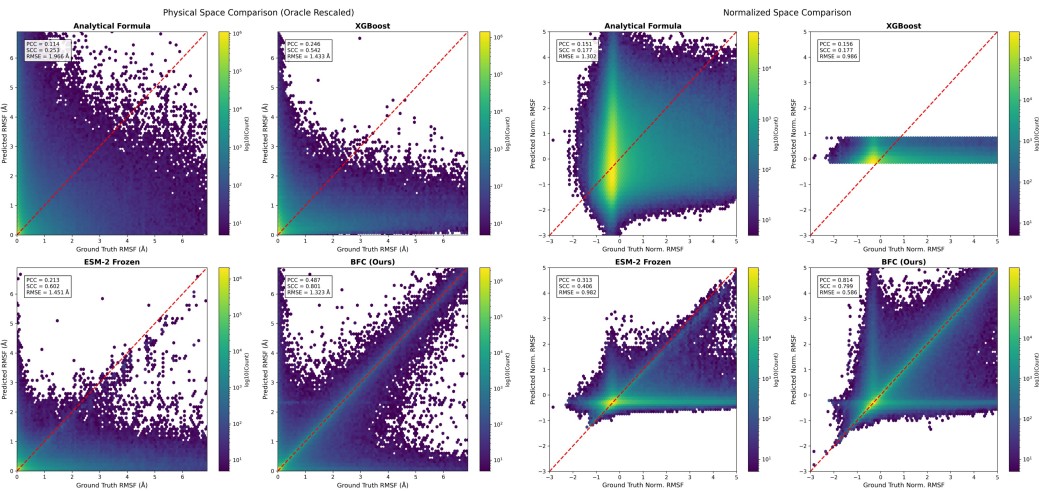

Figure S4: **Comprehensive scatter plot analysis of model performance in physical and normalized spaces.** This figure presents a density-based comparison of predicted flexibility versus ground truth RMSF across four different methods. The panels are organized into two groups: the Physical Space Comparison on the left measures fluctuations in Angstroms, while the Normalized Space Comparison on the right evaluates performance using Z-scores. The color gradient represents the logarithmic density of data points, where yellow indicates high-concentration regions and purple indicates low density. The red dashed line marks the ideal identity relationship where the prediction equals the ground truth. Baseline methods such as the Analytical Formula and XGBoost exhibit significant dispersion and poor correlation. Notably, in the Physical Space Comparison (left columns), all predictions were subjected to Oracle Rescaling (matched to ground-truth mean/std) to ensure a fair comparison of profile shapes. Despite this, the analytical formula shows broad scatter (SCC=0.25), confirming that B-factors are locally distorted. While the Frozen ESM-2 model shows improved correlation, it still retains considerable variance around the diagonal. In contrast, the BFC model demonstrates superior convergence along the identity line in both evaluation metrics. The statistical indicators provided in each panel confirm that BFC achieves the highest Pearson Correlation Coefficient (PCC) and Spearman Correlation Coefficient (SCC) combined with the lowest Root Mean Square Error (RMSE), validating its robustness across different feature spaces.

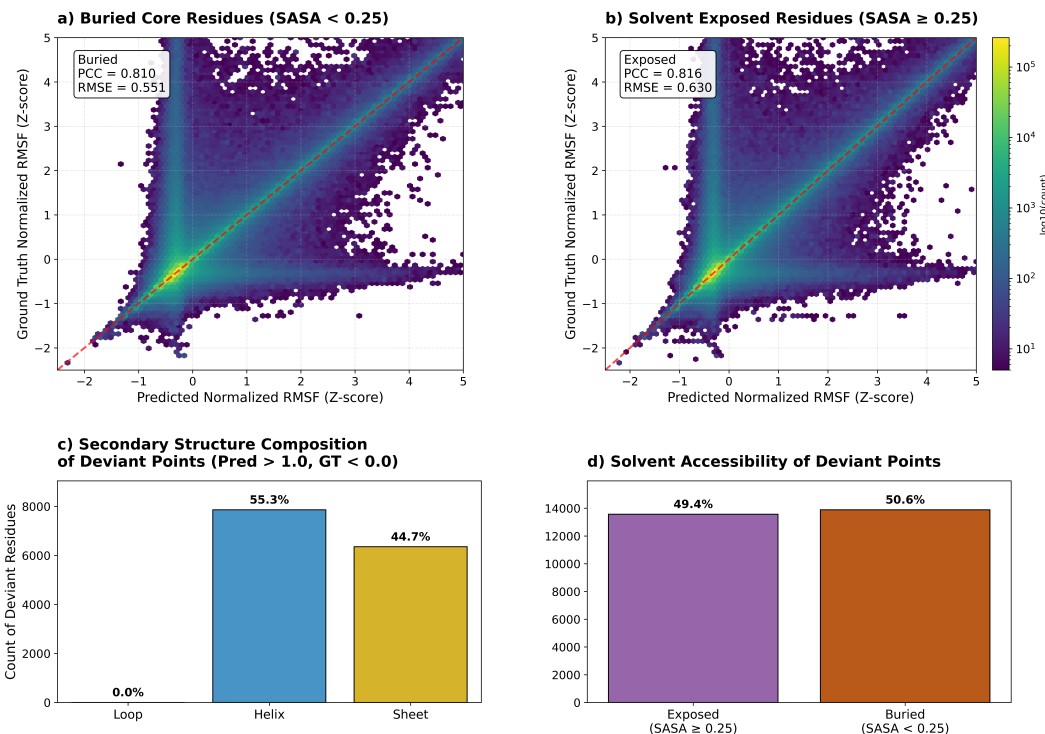

Figure S5: **Stratified error analysis reveals BFC decodes latent flexibility in structurally metastable regions. A, B,** Density scatter plots of prediction performance stratified by solvent accessibility indicate that the model maintains high fidelity in both environments despite distinct baseline fluctuation regimes. The buried residues in **A** exhibit tighter convergence consistent with deterministic packing constraints, whereas the increased dispersion for solvent-exposed residues in **B** aligns with the intrinsic stochasticity of surface loops. **C, D,** Analysis of prediction outliers where the model infers high flexibility (Z-score $> 1.0$) for residues displaying rigid ground truth signals (Z-score $< 0.0$). **C,** Secondary structure composition of these outliers reveals a complete absence of loops and an exclusive distribution among helices and beta-sheets. This pattern suggests the model detects metastable secondary structures that remain folded within the crystal lattice but contain high sequence propensities for local unfolding or conformational exchange. **D,** The balanced distribution of these outliers between exposed and buried residues confirms that these signals arise from intrinsic local frustrations detected by the language model rather than surface artifacts.

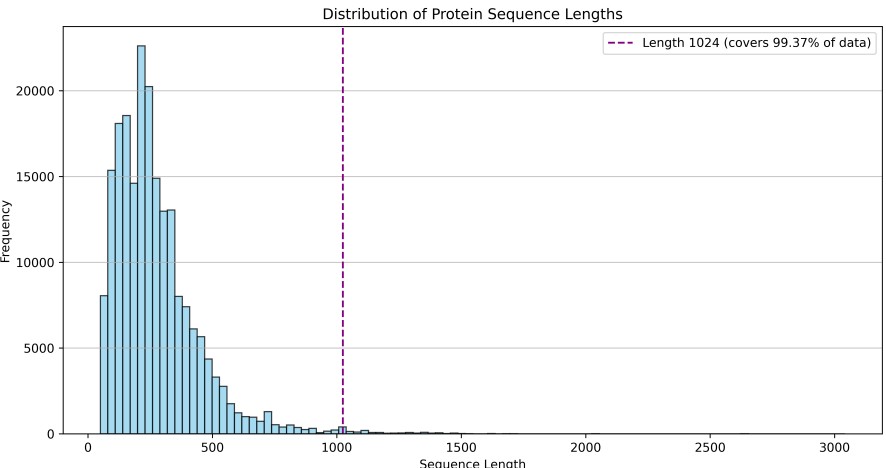

Figure S6: **Distribution of protein sequence lengths.** The vertical dashed line marks the truncation threshold of 1,024 residues set for training and testing. This cutoff covers 99.37% of the total dataset (206,168 out of 207,465 sequences have a length $\leq$ 1,024), effectively reducing computational overhead while preserving the vast majority of biological data.

