# OpenReview forum: "Sequence-context-aware decoding enables robust reconstruction of protein dynamics from crystallographic B-factors"
_ICLR.cc/2026/Workshop/LMRL — ICLR 2026 Workshop LMRL Poster_

### Official Review · Reviewer_Q28y · 2026-02-11

**Rating:** 6
**Confidence:** 5

**Review:**

The authors train a deep learning architecture containing a pLM to predict normalised per-residue RMSF profiles using sequence and crystallographic B-factors, showing strong correlation with MD-derived fluctuations and demonstrating practical utility in docking and interface refinement.

- The jump to approximately 0.8 PCC on the test set (versus very low correlation for normalised B-factors) is substantial and correctly documented.
- The applications to docking and interface refinement seem quite convincing to me. My only concern here would be the apparent small 30-sample test set for the interface refinement part, which also presents results that are mostly supported by illustrative and energy-based analyses rather than quantitative benchmarking.
- Note that RMSF does not capture e.g. concerted motions and thus does not describe all the protein dynamics.
- The method is presented as recovering intrinsic dynamics masked by crystal packing (‘decoding the dynamics’). However, it is not entirely clear to me to what extent this reflects true physical disentanglement rather than learning typical data-driven statistical mapping (in this case, learning sequence-dependent flexibility patterns from the training distribution).
- While the decision of expecting sequence plus B-factors alone to recover solution-state dynamics is conceptually interesting, I find it as an unrealistic scenario as one typically has the resolved structure as well when B-factors are available. It would be useful to see comparisons against structure-aware baselines, as the current setup would seem to deliberately restrict the input, thereby understating what could be achieved with geometric information.

---

### Official Review · Reviewer_eMde · 2026-02-11
**A method for extracting hidden dynamics from B-factors with a protein language model prior**

**Rating:** 8
**Confidence:** 4

**Review:**

This is an interesting paper that introduces a new method for fine-tuning the protein language model ESM to recover protein dynamics that might be missed by traditional B-factor analysis. At the heart of the paper is the idea that sequence-based features can identify dynamic regions of proteins that may appear rigid from crystallography data due to their position in the lattice.

Pros:
- The idea of incorporating ESM with B-factor data is unique and effective.
- The authors demonstrate interesting biological capabilities, such as correcting crystal packing artifacts, identifying cryptic hinges latent in PDB data, and improving molecular docking.

Cons:
- Several of the figures are difficult to read even on a computer screen due to font size. Moreover, while the method looks highly effective, the figures could highlight this more clearly. For example, the authors could consider reserving the graphical figures for visual comparisons of their method against others, while removing the various bar, line, and box-and-whisker plots, collapsing the important information captured by these plots into single scalar metrics that are arranged in a table and compared against other methods.

---

### Official Review · Reviewer_5R5G · 2026-02-21
**Review for submission 25**

**Rating:** 8
**Confidence:** 4

**Review:**

This paper introduces the B-Factor Corrector (BFC), a fine-tuned ESM-2 language model that treats protein dynamics recovery as a sequence-to-dynamics translation task. By leveraging deep contextual embeddings, BFC successfully decouples intrinsic solution-state flexibility from experimental crystal packing artifacts, achieving a Pearson correlation of 0.80 against ground-truth ensemble profiles. Beyond statistical gains, the model demonstrates high practical utility by identifying disease-critical motifs and unmasking cryptic mechanical hinges. Its integration into molecular docking and antibody refinement protocols further proves its value in resolving steric clashes and identifying binding-competent conformations.

The model's primary strength is its minimalist input strategy, which relies solely on raw sequence and B-factors to bypass the need for hand-crafted structural descriptors or expensive simulations. This approach allows it to identify functional features, such as the RAF1 chaperone hinge, that remain obscured even in standard molecular dynamics (MD) trajectories. However, a limitation is its reliance on oracle rescaling for physical space evaluation, as predicting absolute fluctuation amplitudes remains challenging without knowing global scaling factors. Additionally, while BFC is a highly efficient approximation for high-throughput screening, it is not a replacement for MD when studying complex protein transitions.

---

### Meta-Review · Area_Chair_JqUc · 2026-02-25

**Recommendation:** Accept (Poster)
**Confidence:** 4

**Metareview:**

Accept.

---

### Decision · Program_Chairs · 2026-03-02

**Decision:**

Accept (Poster)

**Comment:**

Please see the meta-review.